# Supporting Neurologic Health with Mushroom Nutrition

**DOI:** 10.3390/nu17091568

**Published:** 2025-05-02

**Authors:** Victoria Bell, Palmen Dimitrov, Tito Fernandes

**Affiliations:** 1Faculty of Pharmacy, University of Coimbra, Polo das Ciências da Saúde, Azinhaga de Stª Comba, 3000-548 Coimbra, Portugal; victoriabell@ff.uc.pt; 2LAQV-REQUIMTE, Portuguese Research Centre for Sustainable Chemistry, Rua D. Manuel II, Apartado, 55142 Porto, Portugal; 3Department of Psychiatry and Medical Psychology, Faculty of Medicine, Medical University of Varna, 55, 9002 Varna, Bulgaria; dr_pdimitrov@ymail.com; 4Centre for Interdisciplinary Research in Animal Health (CIISA), Faculty of Veterinary Medicine, University of Lisbon, Avenida da Universidade Técnica, 1300-477 Lisboa, Portugal

**Keywords:** mental health, depression, neurological disorders, nutrition, mushrooms

## Abstract

Due to the extensive types of etiologies and risks causing over 600 types of mental health issues, to convene adequate recommendations in primary care is a difficult assignment. The starting point for preventive interventions on neurologic disorders involves scrutinizing the risk factors while targeting multiple hazards in order to increase the success of an early precautionary mediation plan of action. The primary risk factor for most neurodegenerative diseases is the increasing worldwide median age, although one in seven youngsters also experience a mental disability, namely depression, representing a decline in well-being and conferring a considerable global public health challenge. The brain operates optimally when supported by a holistic approach engaging several aspects, and diet is becoming an integral part of care strategies. Treatment is presently dominated by pharmacotherapy, but additional strategies are needed to prevent and treat mental disorders. Dietary modification can prove to be a cost-effective strategy for the prevention and, in certain conditions, treatment of neurological disorders. Molecules of dietary ingredients, micronutrients, phytonutrients, and additives may modulate depression associated biomarkers. Nutritional exposure during the early developmental stages and maternal impact, lifestyles, and the modulation of the gut microbiota through diet as novel therapies for the treatment of various neuropsychiatric conditions is gaining interest for maintaining brain health. Bioactive substances present in different mushroom species have been ascribed to both direct and indirect mechanisms of influence on neurobehavior, and here we support the recognition of mushroom nutrition as an influential dietary element in prevention and management of some neurologic concerns. Scientific evidence demonstrating the unequivocal link between nutritional mushrooms and cognitive health is only beginning to emerge, and nutritional medicine should be considered as an integral part of mental care.

## 1. Introduction

The trends and rates of global population aging differ considerably in diverse settings, presently being much prominent in high- and middle-income countries, due to the longer average lifespan, falling birth-rates, and other biological changes, as well as physical and social environments.

Age advancing is linked with the rapidly raised incidence of neurodegenerative diseases; around 14% of the world’s population experience mental disorders. Globally, even some 15% of children and adolescents, due to physical, emotional, and social changes, including exposure to poverty, abuse, or violence, suffer from mental health problems [1,2].

These illnesses represent a considerable reduction of health promotion and protection of people and communities, as they critically compromise cognitive and motor tasks, eventually causing a significant decline in well-being and quality of life, causing stigma and discrimination while placing a substantial strain on healthcare systems worldwide [3,4,5].

As a response and in an effort to integrate psychiatry with the mainstream of medicine, the WHO’s Mental Health Action Plan 2013–2020 was adopted at the 66th World Health Assembly, later extended until 2030 by the 72nd World Health Assembly in May 2019, to ensure its alignment with the 2030 Agenda for Sustainable Development [6].

There are more than 600 different types of nervous system disorders [7] including more than 200 classified types of mental illness. The definition of a mental disorder is an ongoing process and has been addressed in successive revisions over time. A mental disorder is defined by its significant clinical impact on an individual’s emotional control, cognition, ideas, frame of mind, conduct, and attitude, linked with distress in important areas of physical performance. It can last for a short time or for an individual’s whole life [8].

Mental health disorders or harmful dysfunctions include emotional instability (e.g., loss of mind or manic depression), uneasiness, fear, and disarray of character, anorexia, bulimia and binge-eating disorder, schizophrenia, post-traumatic stress disorder, and substance abuse disorders [9,10].

The International Classification of Diseases (ICD-11) released by WHO in 2022 is now officially in effect for the multilingual and international digital recording and reporting of causes of illness and more, containing around 17,000 unique codes, more than 120,000 codable terms, even traditional medicine, and presently interpreting more than 1.6 million terms [11].

Mental health disorders continue to rise globally and in 2024, one of every eight people (970 million people), 84 million people in the European Union (1 in 6), were living with a cerebral disability [12]. The widespread of these issues included significant disturbances in thinking, emotional regulation, or behavior, with anxiety and depressive disorders being the most common. Indeed, depression strikes some 280 million people worldwide, and anxiety disorders impact in excess of 300 million individuals [13,14].

Huge numbers and types of risk factors may cause mental illness. Therefore, it is necessary to identify as many risk and protective factors that may influence individuals at different stages of development as early as possible to identify opportunities for effective intervention [15,16].

Prevention is a concept as old as medical practice itself. However, only recently has it become generally accepted in mental health [17]. The starting point for preventive interventions on mental disorders involves scrutinizing sharply and ethically the risk and protective factors, since not all of the evidence from a hazard assessment will be conclusive enough to validate the design of a preventive intervention [18]. Mental disorders, including depression, anxiety, and bipolar disorder, account for a significant proportion of global disability and pose a substantial social, economic, and health burden [19].

Here, we support the recognition of foods and diet, namely of mushroom nutrition, as crucial elements in the prevention and management of neurologic disorders [20]. Nutraceutical approaches, as an adjuvant strategy of a personalized nutritional procedure based on dietary bioactive compounds, are emerging as primary strong match for their neural recovery, regeneration, and therapeutic roles, addressed to improve cognitive dysfunction associated with brain disease states, such as Parkinson’s Disease, Alzheimer’s disease, Multiple Sclerosis, anxiety, and depression [21,22,23,24].

## 2. Mental Health

Mental health is more than the absence of mental disorders. It enables people to cope with the hardships of life and develop their talents. It is a basic human right, and what constitutes it varies for each person [25]. Main determinants of mental health include earnings, occupation, socioeconomic situation, level of schooling, food security, dwellings, social integration, discrimination, childhood misfortunes, the physical, environmental, and societal conditions in which people live, and the ability to access fair and reasonable health care [26,27].

The mental health of people with psychiatric disabilities and illnesses, emotional disorders, or intellectual impairments comprises various frames of mind and well-being that enable people to cope with stress of varying degrees. The intricate interplay of biological, social, and psychological factors determines the level of mental health of a person at any point in time [28,29,30].

Regardless of the specific mechanism of action and causal pathways, alternative remedies represent another option for treating mental conditions, and developing data suggest that bioactive compounds from several natural organic products conceal properties that preserve neuronal structures and/or functions. These products act in the brain, targeting oxidative stress, neuroinflammation, and neurodegeneration [31,32] (Figure 1).

It must be emphasized that not all mushrooms have the same neuroprotective effects. Some have been known for their toxicity for millennia, and major distinctions must also be made between the extract and biomass dietary preparation forms [33].

## 3. Gut Microbiota Involvement in Neurological Diseases

The gut microbiota comprises some 40 trillion microorganisms belonging to over 3000 species, including bacteria, fungi, and viruses [34], and it is estimated that 400–500 different genera of gut microbiota make up the human intestinal dynamic environment ecosystem, 90% of which are predominantly anaerobic [35,36]. Most of them belong to two key phyla, *Bacteroidota* (synonym *Bacteroidetes*) and *Bacillota* (synonym *Firmicutes*), while other less common groups of microorganisms include *Pseudomonadota*, *Actinomycetota*, *Fusobacteria,* and *Verrucomicrobia* [37].

The microbiota profile is negatively influenced by the intake of ultra-processed foods, which greatly reduce the benign versus adverse gut bacteria that can encourage increased intestinal mucosal permeability, which likewise prompts a cytokine storm, an amplified immune response activated longer than it should be, which in turn spawns chronic neuroinflammation, a major cause of mental health conditions and psychiatric disability [38,39].

The bidirectional communication between the gut microbiome and the brain takes place via diverse pathways, including through chronic inflammation that leads to β-amyloid plaque formation in the intestinal tract and spreading to the brain via the vagus nerve [40,41]. This axis has been shown to influence neurotransmission and the behaviors that are often associated with neuropsychiatric conditions, the immune system, neuroendocrine pathways, and bacteria-derived metabolites [42,43].

Therefore, research targeting the modulation of this gut microbiota as a novel therapy for various neuropsychiatric conditions is gaining interest. The microbiota are expected to play a role in nutritional interventions, including mushroom supplements for supporting brain health [44,45]. Indeed, the manipulation of lifestyle factors such as dietary interventions may represent a successful therapeutic approach to maintain and preserve brain health throughout an individual’s lifespan [46].

Nutritional exposure during early developmental stages may induce susceptibility to the later development of human diseases via interactions with the microbiome, including alterations in the brain function and behavior of offspring, as explained by the gut–brain axis theory [47]. There are also implications of maternal nutrition on neurodevelopmental disorders and the establishment and maturation of gut microbiota in the offspring [48].

In the gastrointestinal tract, there are nerve cells that together mass nearly half as much as in the central nervous system [49]. Bioactivity in the intestinal microbiota is also directly related to normal neuropsychiatric development, affecting gut pathophysiology and central nervous system functions by modulating the signaling pathways of the microbiota–gut–brain axis [50,51,52].

Several investigations of the dynamic microbial system and genetic–environmental interactions with the gut microbiota have shown that changes in the composition, diversity, and/or functions of gut microbes (e.g., gut dysbiosis) affect neuropsychiatric health by inducing alterations in the signaling pathways of the axis [53,54,55].

Communication along the microbiota–gut–brain axis may suffer dysregulation, and there is evidence that the lack of balance of various intermediate or end products of microbial digestion and the broad range of natural neuroactive compounds may initiate various neurodevelopmental and neurodegenerative diseases [56].

The axis of communication primarily acts through neuroendocrine, neuroimmune, metabolic, and autonomic nervous system mechanisms [57]. The gut microbiota interact with the host brain, and its modulations play a critical role in the pathology of neuropsychiatric disorders [58]. Dysregulation of this axis modulates the host’s homeostasis by disrupting the integrity of the intestinal and blood–brain barrier, which protects the central nervous system from pathogens and toxins in the blood, the two layer mucus barrier, and the regulation of brain function and the host’s immunity [59].

Evidence gathered in preclinical and clinical studies shows a positive correlation between gut dysbiosis and the pathogenesis and progression of neuropsychiatric disorders. Long-term dysbiosis leads to overstimulation of neuroimmune system and the hypothalamic–pituitary–adrenocortical axis, along with neurotransmitter imbalance. The aforementioned leads to dysregulation of the cell signaling necessary for maintaining proper bodily functions, contributing to inflammation, escalated oxidative stress, mitochondrial cytopathy, and irreversible neuronal apoptosis [53,60].

## 4. Improvement of Brain Function

The brain is the most critical organ in the body and controls our thoughts, emotions, memories, vision, body temperature, skills, several processes, and behaviors. Prime performance of the brain is crucial to individual health and well-being but is still a topic of considerable debate [61]. The brain operates optimally when supported by a holistic approach engaging several aspects such as a healthy diet, physical activity, enough sleep, mental stimulation, reduced stress, good mood, and social activity [62].

Psychological stress can induce both oxidative stress and cellular stress together with inflammation, which can accelerate the aging process, while the mitochondria, the cellular powerhouses, play a critical role in both acute and chronic stress [63].

Brain development is a complex process, whose rate can evolve through the life cycle. In addition, endogenous gut hormones, neuropeptides, neurotransmitters, and the gut microbiota are affected directly by the composition of the diet [44,64,65,66].

Lipids are the most abundant but poorly explored components of the human brain. The role of poly unsaturated fatty acids (PUFAs) in the regulation of neuroinflammatory processes is crucial, since the brain uses more energy than any other human organ and lipids represent 78% of the dry weight of axon myelin sheaths and 35–40% of the neuron-rich gray matter [67,68].

As a consequence, adequate dietary intake and nutritional balance, through several gut hormones or peptides, have a recognized impact in shaping synaptic resilience, cognitive capacity, and brain evolution, affecting a range of mental activities such as learning, memory, problem-solving, decision-making, feelings, attitude, and the communication between the nervous and the endocrine systems, with consequences on health [69,70].

Diets rich in saturated fats and sugar may impair brain performance [71]. Anxiety and depression are diseases increasingly present today, considered the diseases of the century, drastically affecting the quality of life of the population, and elevating the risk of developing chronic diseases [72].

The human brain is metabolically expensive, using a substantial portion (20–25%) of the body’s total energy and nutrient intake, related to glucose metabolism for synaptogenesis, neuronal communication, and knowledge organization [73]. Nutrients strongly influence brain structure and function, neurodevelopment, and neurotrophic function [74]. It has been recognized that diet and nutrition may be an important factor contributing to psychiatric morbidity, and that prevention or treatment of psychiatric disorders could be conducted by addressing diet and nutrition [75,76].

Neuraminidase 1 increases microglial phagocytosis and sensitizes neurons to glutamate, thus potentiating neuronal death, suggesting that neuraminidase 1 might be a possible therapeutic objective to safeguard against neuroinflammatory harm to neurons [77,78,79]. Neuraminidase inhibitors from functional mushrooms have been noted to support neurogenesis and nerve health [80].

## 5. Diet and Nutrition Impact on Mental Health

The diversity of drugs provided to treat neurodegenerative diseases is limited. More recent, research expanded on the effects of nutrition on mental condition, representing an easy and affordable cornerstone for the prevention and reduction in the incidence of many mental disorders [81,82].

It is becoming evident that several bioactive compounds and secondary metabolites from mushrooms and several other foods (e.g., green tea) contain neuroprotective determinants with antioxidant and anti-inflammatory effects. Neurotrophic factors are naturally occurring proteins which are essential for neuronal survival and differentiation during development.

Some food bioactive components can modulate key pathways such as the intracellular phosphatidylinositol 3-kinase/protein kinase B (PI3K/AKT) and brain-derived neurotrophic factor-tropomyosin receptor kinase B-cAMP response element-binding protein (BDNF-TrkB-CREB), which are key to neuronal viability [4,83,84,85] (Figure 2).

Mushrooms themselves, namely from phylum Ascomycota, do not produce matrixins, but have been shown to inhibit them [86].

Indeed, recent data [87] reveal a strong link between the BDNF/TrkB system and the intellectual disability in stress-associated disorders, and that neurotrophins, in particular BDNF, are key molecules which are essential for neuroprotection, cell specialization, and the up-regulation of synaptic plasticity [88,89]. The therapeutic potential of BDNF mimetics in Alzheimer’s (AD), Parkinson’s Disease (PD), and Huntington’s disease (HD) remains in the preclinical stage [90].

Modern scientific research is clarifying the food–mind bidirectional connection and the profound link between dietary choices and brain health [91]. As it happens, there are general clinician guidelines to globally inform psychiatric/medical and health professional practitioners on the use of nutraceuticals and phytoceuticals, on their safety and tolerability, aimed at providing an evidence-informed approach to assist clinicians, service users, and stakeholders in determining rules on the use of such natural agents, across major psychiatric disorders [92,93].

This new rapidly emerging field of nutrition and mental health, coined by leaders in the field as “nutritional psychiatry”, as a crucial factor in the high prevalence and incidence of mental disorders, suggests that food and diet are as essential to psychotherapy as they are in gastroenterology, cardiology, and endocrinology [94,95].

There are prevailing opinions about the health effects of certain foods that can significantly influence mental health and well-being, but the evidence for many diets is comparatively weak and not supported by solid evidence and clinical scientific verification. Therefore, the unequivocal link between nutrition and mental health is only beginning to emerge [96,97].

Although there are some disorders for which this connection between diet and mood disorders, anxiety, and depression is firmly established [98], current epidemiological data on nutrition and mental health do not provide full information about causality or underlying mechanisms [99].

Dietary type (e.g., a Mediterranean diet) can reduce the risk and prove to be a cost-effective strategy for the prevention and treatment of depression among adolescents [100]. Indeed, diet and nutrition need to be recognized as key modifiable targets for the prevention of mental disorders, and nutritional medicine should be considered as an integral part of psychiatric treatment [101,102,103].

Molecules of dietary ingredients, micronutrients, phytonutrients, and additives may modulate depression associated biomarkers [104]. In this context, several healthy foods such as berries, nuts, cruciferous vegetables, non-processed oatmeal, leafy greens, beans, mushrooms, olive oil, fish, legumes, dairy products, fruits, and some spices have been inversely associated with the risk of depression and may also improve symptoms [105].

As an example, research shows that increasing essential fatty acids have an effect on the prevention and treatment of anxiety and depression. Additionally, omega-3 (e.g., in fish, chia seeds, walnuts, seeds, seaweed, olive oil) can optimize mood stability and cognitive function even in young people [106].

Conversely, western dietary patterns, including the consumption of highly processed foods, pre-packaged foods, refined grains, processed and red meats, sugary and syrupy beverages, fried foods, baked products, have been shown to be associated with an increased risk of depression in longitudinal studies [107].

A diet that lacks the necessary micronutrients and bioactive compounds that contribute to the normal, non-pathological functioning of the organism, eaten for an undefined period, can lead to the development of mental health disorders, especially anxiety or mood disorders such as depression and/or increased levels of stress [108,109,110].

Natural bioactive ingredients from mushrooms, microalgae, plants, and cyanobacteria have been deeply investigated for their preventive or therapeutic potential [31,111].

Diet is an important modifiable risk factor for Alzheimer’s disease and dementia, as it is able to modulate structural brain connectivity, cause positive changes in brain function and behavior, and help regulate cognition and emotion [112,113].

However, there appears to be a general belief that dietary advice for mental health is framed around a base of still unsound scientific evidence regarding shaping the brain’s metabolism and impacting the trajectory of various neurodegenerative diseases [64,114]. In reality, for many such claims, it has been very difficult to prove that particular foods and beverages or specific dietary components contribute to or hinder mental health by causing, preventing, or treating disease [92,99,115].

## 6. The Role of Mushrooms on Neurologic Health

This concise review focuses on the latest scientific evidence supporting the link between diet and mental health, with a specific interest in the neuroprotective aptitude of mushroom nutrition and therapies that have psycho-protective abilities significant from the point of view of public health.

There are more than 50 different kinds of macro- and micronutrients in foods. Plant foods contain over 25,000 phytonutrients with bioactive compounds [116,117]. Mushrooms, as macro fungi constitute a dynamic source of unique micronutrients missing in foods of plant or animal origin. They are considered vital functional foods and utilized for the prevention of numerous diseases [118].

The bioactive phytochemical constituents of mushrooms vary profoundly with the species in question, and their bioactivity depends on cultivation conditions, processing methods, and processing techniques [119,120,121,122].

In developed countries, anxiety with disturbing feelings of distress, agitation, fear, and depression with persisting unhappiness, melancholy, or desperate mood are among the most common primary care service challenges in medical healthcare [123].

Unlike the common mushrooms found in grocery stores (e.g., button, shiitake, cremini, Portobello, oyster, enoki, porcini), medicinal functional mushrooms like Lion’s Mane (*Hericium erinaceus*), Reishi (*Ganoderma lucidum*), Cordyceps (*Ophiocordyceps sinensis*), Trametes (*Coriolus versicolor*), Shiitake (*Lentinula edodes*), Chaga (*Inonotus obliquus*), and oyster mushroom (*Pleurotus ostreatus*), among others, contain bioactive compounds that can influence brain health, immunity, and overall well-being [119,124,125] (Table 1).

While chronic neurodegenerative diseases are complex, and their pathogenesis is still uncertain, it has been inferred that brain insults (e.g., by virus infections, HIV, or herpes) may also activate retro-transposons and silent human endogenous retroviruses sequences (ERVs), which constitute up to 8% of the human genome, contributing to neurodegenerative mechanisms [126]. Chronic ERV activation may cause progressive neurodegeneration in genetically susceptible people thereafter, consolidating in cognitive impairment and Alzheimer’s disease/dementia [127,128,129,130].

### 6.1. Lentinula Edodes (Berk.) Pegler (“Shiitake”)

Substantial research has been conducted on polysaccharides from *Lentinula edodes* (Shiitake), one of the most cultured and consumed mushrooms. Shiitake was shown to prevent cognitive impairments associated with obesity, mainly by modulating the gut microbiota community [131].

### 6.2. Grifola Frondosa (Dicks.) S.F.Gray (“Maitake”)

Maitake is a lignin- and cellulose-degrading basidiomycete, an edible wood-decay mushroom, and the intervention effects of their dietary polysaccharides on Alzheimer’s disease (AD), Parkinson’s Disease (PD), depression, anxiety disorders, autism spectrum disorder, epilepsy, and stroke has been investigated among several mushroom sources [132,133].

### 6.3. Trametes Versicolor (L.) C.G. Lloyd) Coriolus Versicolor (“Turkey Tail”)

The immunodulation activity of mushroom biomass of *Coriolus vesicular* (Turkey Tail) in neuroinflammatory pathogenesis with the consequent endogenous cellular defense mechanism modulation and neurohormesis reflects the activation of LXA4 signaling and modulation of stress responsive vitagenes encoding for heat shock proteins. This could serve as a potential innovative treatment for AD-related inflammasome and progressive neurodegeneration [134].

The mushroom *Coriolus versicolor* (Turkey Tail), chronically administered as a dietary supplement approach, showed neuroprotective potential by helping in the response to oxidative stress and acting on the α-synuclein neuronal protein, on transcription factor NF-kB-mediated inflammatory response, and the immune system in Parkinson’s Disease [135,136,137].

### 6.4. Inonotus Obliquus (Fr.) Pilát. “Chaga”

Chaga mushroom (Inonotus obliquus), is often known as “the king of medicinal mushrooms”. It is not a true mushroom, but a hard compact mass of mycelia (sclerotia), ready to survive hostile environmental circumstances [138]. This wild edible mushroom, being rich in various antioxidants, boosts immunity, improves brain and liver health, increases life span, and may thwart the development of Alzheimer’s disease [139,140,141].

### 6.5. Hericium Erinaceus (Bull.:Fr.) Pers. (“Lion’s Mane”)

In Australia, preclinical testing found *Hericium erinaceus* (Lion’s Mane) mushrooms to have a substantial effect on the growth of brain cells and improving memory [142]. Studies revealed that this mushroom contains approximately 150 small molecules, the two most well-known categories being hericenones and erinacines, which can stimulate the growth of brain neurons and glia cells [143]. These active compounds, plus polysaccharides, steroids, alkaloids, and lactones, can help promote neurogenesis and enhance memory [144], having potential beneficial effects in ameliorating cognitive functioning, and behavioral deficits in AD and PD [145,146].

The traditional medicinal mushroom *Hericium erinaceus* is also known for enhancing peripheral nerve regeneration through targeting nerve growth factor (NGF) neurotrophic activity [142,146,147]. It was demonstrated that this mushroom is effective in improving mild cognitive impairment in older people [148,149]. Preclinical studies have shown that there can be improvements in ischemic stroke, PD, AD, and depression if *Hericium erinaceus* mycelia enriched with diterpenoid erinacines are included in daily meals [150,151,152].

The main question has been the dosage to be taken, and universal guidelines have not yet been set for Lion’s Mane intake. Taking up to 1 g orally every day for up to 16 weeks may be safe and well-tolerated. The observations cautiously indicate that *Hericium erinaceus* may improve performance celerity and minimize the experience of distress in healthy young adults. However, limited and gloomy findings were also recorded [153]. Due to reduced sampling size, the encouraging results should be treated with prudence [149,154]. Medicinal mushroom incorporation in diets, at least twice a week, was shown to reduce the risk of the early stage of memory loss, usually anticipating neurological diseases [155,156].

*Hericium erinaceus* also revealed an inhibitory activity against some bacteria responsible for triggering multiple sclerosis, a chronic neurological disorder, and other selected autoimmune diseases [157].

### 6.6. Ophiocordyceps Sinensis (Berk.). Cordyceps (“Caterpillar”)

*Cordyceps*, a fungus that lives on certain caterpillars, produces promising bioactive metabolites, like β-glucans, adenosine, cordycepin, and ergosterol. Although a significant number of bioactive elements is known from *Cordyceps*, only a handful have been assessed for their neuroprotective ability. The literature still lacks information from clinical trials [158].

In Asia, *Cordyceps militaris* is a medicinal mushroom traditionally used in tonics for treating several neurological disorders, including epilepsy and anxiety. Reports have shown that this mushroom has anti-inflammatory and anti-oxidative effects and may be beneficial for depression management [159]. However, the pharmacodynamics of *Cordyceps* admit a brief lifetime and reduced digestibility, which restricts the treating or mitigating capacity and outcome [158].

### 6.7. Ganoderma. Lucidum (Curtis) P. Karst (“Reishi”)

The Reishi mushroom (*Ganoderma lucidum*) has long been known for its benefits on the mind and emotions, and recent research reveals its ability, to heal and restore the structure of axons and dendrites by modulating neurotransmission, neuroplasticity, and maintaining redox homeostasis, improving the electrical and chemical signals of neurons within the nervous system [160,161].

The *Ganoderma* genus has over 300 different species, and in *Ganoderma lucidum* ganoderic acid A is one of the major triterpenoids. Accumulating evidence has indicated that *G. lucidum* and its several hundred secondary metabolites (especially triterpenes and aromatic meroterpenoids) demonstrate several biological target impacts and displays healing power for diverse neurological disorders [162,163,164,165].

### 6.8. Pleurotus Eryngii (DC. ex Fr.) Quel. (“King Oyster”)

Some antioxidant peptides and protein hydrolysates derived from gastrointestinal digestion of the King Oyster mushroom, *Pleurotus geesteranus*, have been the source of bioactive molecules in the prevention, relief, and even treatment of neurodegenerative disorders [166,167].

Secondary metabolites from natural compounds such as mushrooms (e.g., flavonoids, terpenes, phenols, alkaloids, and polysaccharides) show neuroprotective properties and may be grouped according to their bioactivity as for: (a) preservation of cognitive functions; (b) AChE (anti-cholinesterases) inhibition; (c) anti-neuroinflammation; (d) anti-apoptotic; (e) anti-amyloidogenic; (f) and autophagic stimulation [31].

Emerging nutraceuticals are showing promise as modulators of mitochondrial redox metabolism, capable of eliciting beneficial outcomes. Mushrooms, known for their potent antioxidant properties, have attracted interest due to their potential neuroprotective, antioxidant, and anti-inflammatory effects on mitochondrial dysfunction-associated disorders [168,169]. Therefore, mushrooms can be considered as useful therapeutic agents in the management and/or treatment of neurodegenerative diseases [170,171,172].

The mode of action of mushroom biomass and extracts on the immune system and health has been described, but research is still required, including probing studies in humans to understand the implications of the observed effects on immune function, gut microbiota, cognition, periodontitis, cancer mechanisms, and body weight and composition. Usually, mushroom extracts lack many of the functional enzymes and compounds found in the biomass of same mushrooms [173,174].

### 6.9. Mixture Hericium Erinaceus and Coriolus Versicolor

The mixture of *Hericium erinaceus* and *Coriolus versicolor* biomasses has been the topic of investigation on the regulation of a hormetic-dependent activation of vitagenes, some major antioxidant enzymes, namely, thioredoxin systems, glutathione peroxidase (GPx), superoxide dismutase (SOD), catalase, lipoxin A4, and sirtuins, as potential target G protein-coupled receptors (GPCRs) to treat neurological diseases [175]. The action of this mushroom blend also represents a promising nutraceutical choice for preventing Parkinson’s Disease by acting on neuroinflammation and thus preventing dopaminergic neurons from undergoing apoptosis [176].

A wide range of edible mushrooms have been reported to produce different antioxidant bioactive compounds such as phenolics, flavonoids, terpenoids, lectins, lentinan, galactomannan, glycoproteins, vitamins, carotenoids, ergothioneine, and many others [177], which might be used for dietary supplementation to enhance antioxidant defenses and, consequently, the prevention of age-related neurological diseases [178,179,180].

These compounds have demonstrated several mechanisms, revealing reduction of the pathology of amyloid-beta peptide and microtubule-binding protein tau, neuroinflammation, neuronal apoptosis, and oxidative stress, all responsible for Alzheimer’s disease [181]. They may also act by intensifying the synthesis and the release of acetylcholine (ACh) by inhibiting acetyl cholinesterase activity, inducing an increase in acetylcholine in the synapse [180,182,183].

Several nutrients in mushrooms help to maintain a healthy immune system. In addition, edible mushrooms contain a naturally occurring amino acid, 5-hydroxy-L-tryptophan (5-HTP), which is a primary forerunner of serotonin—a neurotransmitter used in pharmacotherapy of clinical depression [119,184].

Mushrooms also contain organic germanium [185] which is known to have the potential to protect nerve cells and promote their survival and repair [186,187].

Mushrooms are rich in polyphenols (e.g., phenolic acids, flavonoids, tannins), recognized as natural antioxidants [188] that can disrupt the nuclear transcription factor kappa B (NF-κB) pathway, which modulates various features of innate and adaptive immune functions and functions as a critical buffer of inflammatory damage, modulating neuronal survival and further preventing the degradation of IκB, a kinase enzyme complex that is involved in propagating the cellular response to inflammation [189,190,191].

Certain mushrooms have a long history of offering a natural and holistic approach to mental well-being by enhancing cognitive function and resilience by promoting emotional stability that performs the reciprocity of mind, body, and spirit [192,193].

Ménière’s disease (MD) is a cochlear neurodegenerative disease, but despite considerable research, the etiology and pathogenesis of MD remains controversial and undefined. Although usually associated with allergic, genetic, or trauma sources, and with viral infections and/or immune system-mediated mechanisms, it has been found that chronic supplementation with *Coriolus versicolor* mushroom biomass may have a significant impact on the neurotoxic insult and prolonged pro-inflammatory and oxidative status operating in MD pathogenesis [20,174].

Autism spectrum disorder (ASD), a condition related to neurodevelopment occurring in the first 3 years of life, is a heterogeneous group of complex neurological disorders associated with disturbed redox homeostasis [194]. The intestinal dysfunction present in approximately 50% of cases may be characterized by an increase in inflammation, and its treatment has been tested with the presence of natural polyphenols from functional mushrooms in preclinical and clinical trials [195,196].

Edible mushrooms can improve cognitive function and memory [197] and may be considered essential for preventing several age-based neuronal dysfunctions such as Parkinson’s and Alzheimer’s diseases, highlighting their potential to preserve healthy aging by counteracting neurodegenerative diseases [170,180,198,199].

## 7. Summary on the Mode of Actions of Mushrooms on Neuroprotection

The administration of mushroom supplements showed a reduction and prevention on the degree of damage in various brain/nerve tissue conditions and injuries [143] (Figure 3).

Overall, different and possibly cumulative mechanisms of action of several mushroom products may be involved in the preservation of neuronal structure and function and improvement of cognitive functions, namely: (a) improvement of immune function through β-glucans and enzymes (e.g., laccase; SOD-super oxide dismutase, GSH-gluthatione reductase, peroxides, cytochrome P-450 reductase; (b) reduction of inflammation by modulating the inflammasome and increasing lipoxin A4 expression; (c) improvement of the microbiota balance; (d) stimulation of neurogenesis through the production of nerve growth factor (NGF) and increase of the neurogenic reserve; (e) reduction of oxidative stress and cellular stress response by increasing the expression of the proteins thioredoxin and heme oxygenase; (f) preventing mitochondria dysfunction and apoptosis in nerve cells.

The potential of mushroom preparations as functional foods, nutraceuticals, and supportive treatments accentuates the need for integrative work between researchers, clinicians, and regulatory bodies [200].

## 8. Concluding Remarks

Despite centuries of mushroom use in Asia, their vast therapeutic properties are still not fully understood and admissible in the West. Detailed mechanisms of the effects of these functional foods on the human organism still require further long-term clinical studies to confirm their safety of use, tolerability, indications, interactions, dosage, and, in particular, their specific impact on mental health.

Based on the current preclinical data, a great variety of functional foods, nutraceuticals, and phytoceuticals, given either as provisional or supportive recommendations to address a wide array of mental issues, have demonstrated substantial benefits on neurological situations. However, only a few have a frail support for possible immediate application, since in general it was not possible to achieve a clear recommendation instruction, largely due to limited case numbers or mixed study findings.

Human clinical studies require strict ethical approvals and are time consuming, expensive, and often burdensome. It is necessary to recruit the right participants from representative sample groups, collect acute and chronic data using suitable sensitive neurocognitive functioning assessments, and investigate a range of edible functional mushrooms or their dietary supplements to confirm the effects on cognitive function and mood.

Mushroom dietary supplements do not replace current pharmacological treatments for mental health disorders; they should be used in a complementary way. Present data suggest that adhering to a balanced diet combining healthy personalized dietary strategies, including mushroom biomass intake, may reduce the risk of developing neurologic disorders such as depression.

## Figures and Tables

**Figure 1 nutrients-17-01568-f001:**
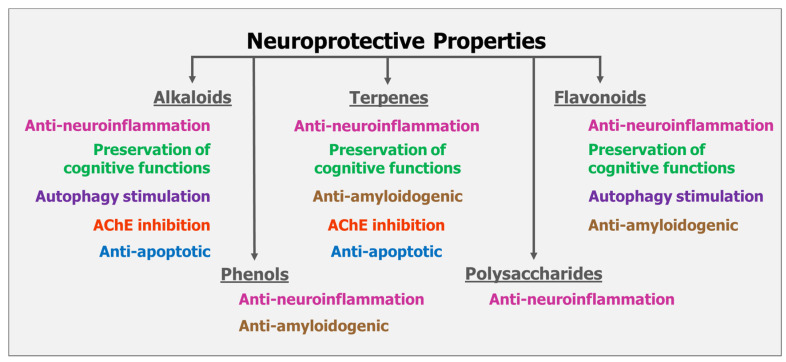
Mushrooms, plants, microalgae, and cyanobacteria contain several classes of natural compounds (i.e., polysaccharides, flavonoids, terpenes, phenols, alkaloids) with neuroprotective properties, that can be organized according to their capacity of interacting with or affecting biological systems (i.e., preservation of cognitive functions, inhibition of acetylcholinesterase (AChE), anti-neuroinflammation, anti-apoptotic, anti-amyloidogenic, and autophagic stimulation).

**Figure 2 nutrients-17-01568-f002:**
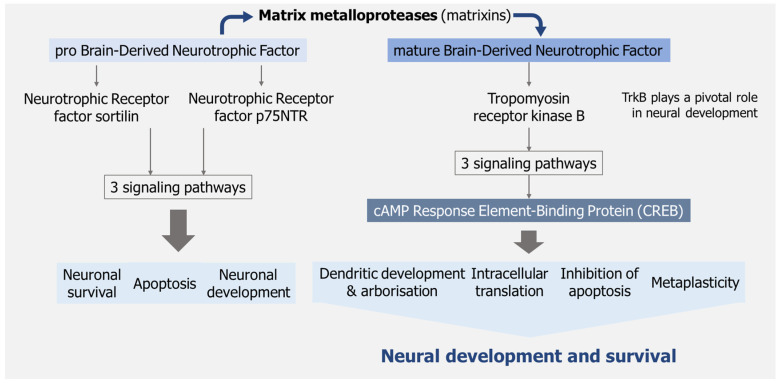
Matrix metallopeptidases (matrixins) are a large family of proteolytic enzymes, calcium-dependent zinc-containing endopeptidases, involved in the inflammatory response, having enormous implications in physiology and disease. The pro-BDNF/p75/sortilin complex leads to promotion processes such as apoptosis, neuronal growth cone development, and neuronal survival. The mBDNF/TrkB receptor complex triggers activation of three signaling pathways that, in turn, activate the transcription factor cAMP Response Element-Binding Protein (CREB) and the genes responsible for development and survival of neurons.

**Figure 3 nutrients-17-01568-f003:**
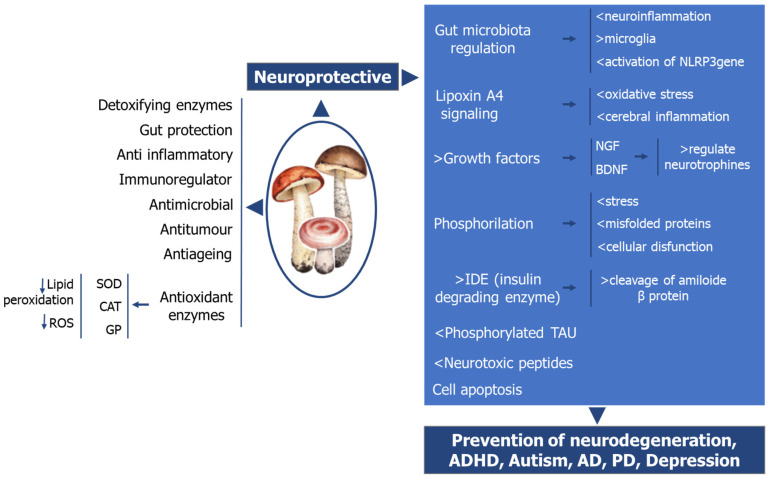
A summary of functional mushrooms neuroprotective activities. PD-Parkinson’s Disease; AD-Alzheimer’s disease. ADHD, Attention deficit hyperactivity disorder. Neuronal NLRP3 gene is an inflammasome multiprotein complex that drives neurodegeneration in PD.

**Table 1 nutrients-17-01568-t001:** Summary table of the genus, species, and common names of medicinal functional mushrooms mentioned in this section. (Authors’ images).

Genus Family	Species Name	Common Name	
Lentinula	*Lentinula edodes*	Shiitake mushroom	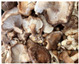
Grifola	*Grifola frondosa*	Maitake mushroom	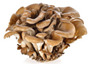
Polypore	*Coriolus versicolor*	Turkey tail mushroom	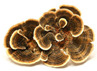
Inonotus	*Inonotus obliquus*	Chaga mushroom	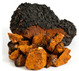
Hericium	*Hericium erinaceus*	Lion’s Mane mushroom	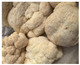
Ophiocordyceps	*Cordyceps sinensis*	Caterpillar mushroom	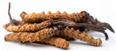
Ganoderma	*Ganoderma lucidum*	Reishi (Lingzhi) mushroom	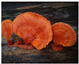
Pleurotus	*Pleurotus ostreatus*	Oyster mushroom	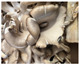

## Data Availability

Not applicable.

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
