# Peer review of "Supporting Neurologic Health with Mushroom Nutrition"

_nutrients, 2025, doi:10.3390/nu17091568_

Round 1
Reviewer 1 Report
Comments and Suggestions for Authors
The title of the manuscript does not really correspond to the content. Whereas the part about mental health is very long the part about the contribution of mushroom nutrition to mental health is short and superficial. A clear relation between both topics is not always recognizable. In the present stage, the sentence in the concluding remarks: „based on current preclinical and clinical data….have demonstrated substantial benefits“ is not enough evidenced.
I recommend describing the results available for the mushrooms in concrete terms (amount of mushrooms consumed, prepared mushrooms or mushroom powder, observed effects, etc.). Perhaps it makes sense to limit the diseases, e.g. to cognitive disorders and depressions.
Minor points:
Line 36: cancel The
44: cancel comma
92: check the sentence
95/96: account, pose
Figure 1: Polysacharides
Fig. 2: intracelular
298: „properties“ is not the right word
322: Health
470: cancel .
475: not all words in italics
481: represents
485: galactomannan and lentinan are polysaccharides
496: it is not correct that all nutrients in mushrooms help to maintaina healthy immune system
- but not in mushrooms
- contains
561: produces
Fig. 3 inflamatory
Page 10 ff.: use the correct names of the fungi and add author names, e.g., Trametes versicolor (L.) Lloyd, Lentinula, Ophiocordyceps
Fig. 1is very common and maybe confusing because only few representatives of the compound classes possess neuroprotective activities, other are, e.g., strongly toxic. This point should become more clearly, at least in the legend.
301, 380: which food bioactive components?
419: 50 kinds….in mushroom foods???
Fig. 3 „summary of active substances“ there are nearly no active compounds mentioned
Comments on the Quality of English Languagesee minor points
Author Response
Please see attachment. The article was edited deleting ( over 2500 words) what the Reviewers considered superfluous. ALL queries and each one was answered on the attached document.

Reviewer 2 Report
Comments and Suggestions for Authors
This manuscript presents a comprehensive review of the emerging scientific evidence linking mushroom-based nutrition with mental health outcomes. The authors explore various neuroprotective, antioxidant, and anti-inflammatory mechanisms by which bioactive compounds in edible and medicinal mushrooms may contribute to preventing or mitigating psychiatric and neurodegenerative disorders.
The manuscript contains overly long sentences and dense phrasing, which hinders readability. I recommend substantial language editing for clarity, flow, and conciseness.
There are several repetitive sections, particularly concerning inflammation and neurotransmitter mechanisms (e.g., BDNF, serotonin), which could be streamlined.
While the topic is broad and ambitious, the review occasionally loses focus, drifting into general dietary or microbiota discussions not directly linked to mushroom-derived compounds.
I suggest sharpening the focus specifically on functional mushrooms and clearly identifying which mental health conditions have the strongest evidence base (e.g., depression, anxiety, Alzheimer's disease).
Many referenced studies are based on animal models or in vitro data. There is a need for more emphasis on human clinical trials and a clearer distinction between well-established findings and preliminary research.
Certain claims (e.g., “counteracts Alzheimer’s”, “improves cognition”) are presented with limited or unclear support. These should be more critically phrased and better referenced.
Line 36, you should deleted "The", due to that it is repeated.
Author Response
We have replied to ALL queries raised by Reviewer 2
Thank you

Round 2
Reviewer 1 Report
Comments and Suggestions for Authors
Please check Table 1 for correct names of the genera (Lentinula, Ophiocordyceps).
Author Response
Dear Reviewer
We made the requested changes in Table 1, referring the names of 2 mushroom families.
Thank you